# Seasonal Trend of Viral Prevalence and Incidence of Febrile Convulsion: A Korea Public Health Data Analysis

**DOI:** 10.3390/children10030529

**Published:** 2023-03-09

**Authors:** Ha Rim Keum, Seon Jin Lee, Jeong Min Kim, Sang Won Kim, Hee Sun Baek, Jun Chul Byun, Yu Kyung Kim, Saeyoon Kim, Jae Min Lee

**Affiliations:** 1Department of Medicine, College of Medicine, Yeungnam University, Daegu 42415, Republic of Korea; dekhr0810@daum.net (H.R.K.); leeejjin7226@naver.com (S.J.L.); aq9497@naver.com (J.M.K.); 2Medical Research Center, College of Medicine, Yeungnam University, Daegu 42415, Republic of Korea; kimsw3767@ynu.ac.kr; 3Department of Pediatrics, College of Medicine, Yeungnam University, Daegu 42415, Republic of Korea; whiteheesun@gmail.com (H.S.B.); sysnow88@hanmail.net (S.K.); 4Department of Pediatrics, School of Medicine, Keimyung University, Daegu 42601, Republic of Korea; junchul1999@hanmail.net; 5Department of Clinical Pathology, School of Medicine, Kyungpook National University, Daegu 41944, Republic of Korea; kimyg@knu.ac.kr

**Keywords:** febrile convulsion, virus, children

## Abstract

Febrile convulsion (FC) is the most common seizure disease in children, which occurs with a fever. We investigated the Korean Health Insurance Review and Assessment Service data of patients aged between 6 months and 5 years at the time of FC diagnosis. Diseases that can cause seizures with fever, such as neoplasms, metabolic disorders, nervous system disorders, cerebrovascular diseases, perinatal problems, and congenital abnormalities, were excluded. Weekly virus-positive detection rate (PDR) data were obtained from the Korea Disease Control and Prevention Agency for adenovirus, parainfluenza virus, respiratory syncytial virus (HRSV), influenza virus, coronavirus (HCoV), rhinovirus (HRV), bocavirus, metapneumovirus (HMPV), rotavirus, norovirus, and astrovirus. Using the Granger test, we then analyzed the monthly PDR and investigated the association between FC incidence and monthly PDR. We additionally identified monthly and seasonal FC incidence trends using the autoregressive integrated moving average. Between 2015 and 2019, 64,291 patients were diagnosed with FC. Annually, the incidence was the highest in May and the lowest in October. Most patients were diagnosed during the spring (26.7%). The PDRs for HRSV, HCoV, HRV, HMPV, and norovirus were associated with FC incidence after 1 month.

## 1. Introduction

Febrile convulsion (FC), the most common cause of seizures during childhood, is defined as a seizure that occurs between the ages of 6 months and 5 years, in the presence of fever (temperature of 38 °C or higher), and which occurs without central nervous system infections (meningitis, encephalitis, and brain abscess), metabolic disturbance, traumatic cause, or history of afebrile seizure [1]. FC occurs in approximately 4% of all children, with a peak occurrence at 12–18 months of age [2].

Simple FC is generalized at onset, lasts < 15 min, and is not recurrent within a 24 h period. Complex FC is more prolonged (>15 min), has focal features, and/or recurs within 24 h. Fever > 40 °C, viral infection, developmental delay, and family history of FC are known risk factors for the first episode [3]. The prognosis of children with simple febrile seizures is excellent, with a low incidence of seizure disorder development and no increased risk for mortality [4] or neurodevelopmental sequelae [5] after a simple febrile seizure.

Viral infections are documented in up to 80% of FC [6], especially in association with upper respiratory tract infections (rhinopharyngitis, pharyngotonsillitis, and otitis media) and gastroenteritis [7]. However, previous studies were mostly institution-based and had some limitations. For example, viral results may be negative in the acute phase at the time of FC diagnosis, forecasting which patient will develop FC is impossible, and virus testing differs from institution to institution. Moreover, there are limited national data on the association between viral infection and FC in Korea. Thus, using the Korean Health Insurance Review and Assessment Service (HIRA) data, this study aimed to identify the virus that causes FC by analyzing the correlation between FC trends and viral epidemics in Korea.

## 2. Methods

### 2.1. Study Population

We extracted data regarding FC from the HIRA, a government-affiliated organization created to build an accurate claims review and quality assessment system for the National Health Insurance, with an open database for all academic investigators [8,9,10]. The claims data in the HIRA database include patient diagnosis, treatment, procedures, surgical history, and prescription drugs, serving as a valuable resource for healthcare service research. Since FC can be accompanied by various diseases, FC in patients diagnosed with other diseases is likely to be caused by the disease. Thus, we studied the HIRA data of FC in patients without diseases that can cause FC, including neoplasms (International Classification of Diseases Tenth Revision [ICD-10], C00–C48); endocrine, nutritional, and metabolic diseases (E00–E90); diseases of the nervous system (G00–G99); certain conditions originating in the perinatal period (P00–P96); congenital malformations, deformations, and chromosomal abnormalities (Q00–Q99); and cerebrovascular diseases (I60–69)]. Patients aged between 6 months and 5 years were analyzed (Figure 1).

### 2.2. Surveillance Data of Virus

We used the data reported by the Korea Disease Control and Prevention Agency (KDCA) on viruses that cause acute respiratory infections and gastroenteritis, where more than 4000 respiratory and 2000 enteric specimens were collected from 17 local environmental and health institutes and over 100 participating hospitals across Korea during each year of the study period. The causative pathogens were then identified using standardized diagnostic procedures in a central laboratory. Pathogen prevalence was surveyed weekly and analyzed based on genetic testing of patients with influenza-like illness or acute diarrhea. The positive detection rate (PDR) data were collected during the study period from 2015 to 2019, calculating the average monthly PDRs of eight respiratory viruses [adenovirus (HAdV), parainfluenza virus (HPIV), respiratory syncytial virus (HRSV), influenza virus (IFV), coronavirus (HCoV), rhinovirus (HRV), bocavirus (HBoV), and metapneumovirus (HMPV)] and four acute diarrhea viruses (rotavirus, norovirus, enteric adenovirus, and astrovirus).

### 2.3. Statistical Analysis

We used the 2015–2019 Korean population data provided by the Korean Statistical Information Service for incidence rate calculations. The age-standardized rate (ASR) was directly adjusted to the 2015 population, with 2,266,781 people under the age of 5. We then constructed a model of variations in FC diagnosis using the autoregressive integrated moving average (ARIMA) modeling approach, which assumes that current observations are related to past observations over time. The general multiplicative form of the ARIMA model is denoted as (p, d, and q), where p, d, and q are the order values of the non-seasonal autoregressive, differencing, and moving average parameters, respectively. Additionally, the autocorrelation function (ACF) was examined to identify the general form of the model to befit. Considering the ACF graphs, different ARIMA models were identified for model selection (Figure 2), and the minimum Akaike information criterion model was chosen as the best-fit model. Table 1 shows the parameters of the ARIMA models in febrile convulsion and the AIC parameters of the model. Moreover, the Granger approach was used to investigate the number of current values in time series y that could be described as other values [11]. The Granger causality test is a statistical technique that analyzes the causal relationship of each time series data, and each time series has an assumption that must follow stationarity. Therefore, it has a normalization process of time series data. At this time, ACF can know whether the time series data are correlated and is used to identify the ARIMA model. Accordingly, the ACF in this study is shown in Figure 2, and Table 1 describes the model and diagnosis of ARIMA. The data were analyzed using R software, and *p* < 0.05 was considered statistically significant.

## 3. Results

### 3.1. Patient Characteristics

During this 5-year-period, 64,291 patients were diagnosed with FC (Table 2). Of the 64,291 patients, 8930 (13.8%) were aged 0–0.9 years, 31,485 (49.0%) were aged 1–1.9 years, 14,641 (22.8%) were aged 2–2.9 years, 6162 (9.6%) were aged 3–3.9 years and 3073 (4.8%) were aged 4–4.9 years. A total of 34,278 males and 30,013 females were included in this study. The average ASR of children under 5 years of age during the study period was 622.4/100,000 population (Table 3). The annual age-standardized incidence rates were 651.5, 683.3, 587.8, 587, and 602.2/100,000 in 2015, 2016, 2017, 2018, and 2019, respectively (Table 3, Figure 3). Based on the age group, the FC incidence rate was the highest in patients aged 1–1.9 years, 1589.0/100,000 person-years, which was 11.9-fold higher than that in children aged 4–4.9 years, followed by 697.9, 492.7, 276.5, and 133.3 in children aged 2–2.9, 0–0.9, 3–3.9, and 4–4.9 years, respectively. (Figure 4). The monthly incidence trends according to the age group were similar.

### 3.2. Trend Analysis of FC

Figure 5A shows the monthly trend analysis of FC from 2015 to 2019. During the study period, the highest incidence was observed from May to July (Figure 5B). The lowest incidence rate in 2015–2016 was from September to November; however, in 2017–2019, it showed a double peak from September to November and February to March. Overall, the average number of cases per month over the period of 5 years was found to be highest in May (1437 cases on average) and lowest in October (739 cases on average) (Appendix A). Moreover, winter (32.3%) was the season in which individuals with FC were identified the most frequently, followed by spring (26.5%), summer (24.3%), and autumn (16.9%). (Figure 6). From 2015 to 2019, there were 12,858 cases annually and an average of 1072 cases monthly.

### 3.3. Positive Detection Rates of Virus

The PDRs of most viruses showed seasonal variations (Appendix A). Specifically, HAdV was the highest in August, HPIV in May, and HRSV from November to December. Moreover, IFV was particularly high during winter, with the highest number of cases in January. HCoV was the highest from December to January, HRV from September to October, and HBoV from May to June. HMPV was the highest in April, rotavirus in March, norovirus in December, enteric HAdV in September, and astrovirus in January. The PDR of all viruses was the highest in December and the lowest in August.

### 3.4. Causality between FC and Virus Prevalence

The prevalence of the virus may rise before the peak of FC diagnosis if any prevalent viruses affect FC diagnosis. Thus, a Granger causality test was carried out between the viral PDR and FC diagnostic information obtained a month later. Table 4 shows the test results. The frequency of several viruses among the eight respiratory and four gastrointestinal viruses elevated one month before the FC incidence. At one month, a higher FC incidence was associated with the PDRs for HRSV in patients under five (*p* < 0.01). After one month, there was a correlation between the PDRs for HCoV (*p* = 0.022, 3–3.99 years; *p* = 0.009, 4–4.99 years), norovirus (*p* = 0.038, 3–3.99 years; *p* = 0.008, 4–4.99 years), and HRV (*p* = 0.039), as well as the PDRs for HMPV (*p* = 0.010) and HMPV in patients aged 1–1.9 years. Figure 7 shows the correlation between the FC incidence during the study period and the PDRs of HRSV, HRV, HCoV, HMPV, and norovirus.

## 4. Discussion

This study evaluated the incidence of FC and its association with a viral infection. FC is a convulsion that occurs with a fever. As viral infection is the most common cause of fever in children, it is necessary to investigate the relationship between FC and viral infection. We found that certain respiratory and gastroenteritis viral PDR were significantly associated with the incidence of FC after 1 month.

There are two major explanations for seizure induction of fever in FC, neural and inflammatory pathways. Regarding neural excitation, it has been speculated that increasing the brain temperature increases the likelihood of synchronized neural activity leading to convulsions [12]. Hyperthermia alone can excite pyramidal and dendritic granule cells in vitro. In addition to the neural pathway, fever generates FC through the inflammatory pathway. In FC rats, interleukin-1c, a pyrogenic cytokine, increased and was maintained at the start of FC. Cytokines differ according to the type of viral infection, and this difference can alter the risk of FC occurrence. Pre-existing brain organic problems may be revealed as seizures caused by inflammatory cytokines.

The prevalence of FC varies geographically and is higher in Asian countries than in Western countries. Prevalence rates were 4.3% in Turkey [13], 0.64% in Brazil [14], 8.3–9.1% in Japan [15,16], and 11.19% in Korea [17]. The same tendency was observed for the FC incidence. The incidence of FC was 2.0% in the United States, 2.3% in Britain [18], 71.1/100,000 person-months (male), 58.9/100,000 person-months (female) in Denmark [19], 10% in India [20], and 0.35% in Hong Kong [21]. According to a study based on Korean national registry data, the prevalence rate under the age of 5 was 6.92–11.19%, peak prevalence was shown in the age group of 2–2.9 years, the incidence was 25.4–65.3/1000, and the peak incidence was shown in the age group of 1–1.9 [17,22,23].

In our study, the overall incidence rate under the age of 5 years between 2015 and 2019 was 622.4/100,000 individuals. The reason for the lower overall incidence compared to the results of previous Korean studies is presumed to be the different study populations. Unlike previous studies that included all patients with FC diagnosis (ICD-10, R56.0), we excluded patients with underlying diagnoses, such as neoplasms, endocrine, nutritional, metabolic diseases, nervous system disease, conditions originating in the perinatal period, congenital malformations, cerebrovascular diseases that can cause seizures with fever.

FC occurs irregularly throughout the year, and the seasonal patterns vary from country to country. In Finland, FC occurs most frequently during winter. Febrile episodes occur most frequently and least frequently during summer. [24] Two peaks in November-January and June-August in Japan [25] and a peak from April to July in Korea [23] were reported. In our study, the incidence was the highest from May to July, like previous studies in Japan and Korea. The peak in summer could be interpreted as an association with a gastrointestinal infection, and the peak in winter may be explained by the upper respiratory tract infection tendency in children.

FC is closely associated with viral infections [12,26]. In Japan, epidemic IFV infection and infectious gastroenteritis are independent risk factors for FC [27]. Viruses, such as IFV A and B, HRSV, HAdV, HMPV, HPIV, HRV, rotavirus, enterovirus, and human herpes virus 6 have been shown to be associated with FC [12,28]. In a multicenter prospective study in eight cities in Turkey, Carman et al. reported that at least one virus was identified in 144 of 174 children (82.7%) [28]. In their study, adenovirus was the most frequently detected virus; IFV was detected in 47.2% (IFVA 24.3% and IFV-B 22.9%), and HRSV in 16%. IFV-B was the most frequently detected virus in the first febrile episode, HRSV-A was the most common virus in simple FC, and human bocavirus was the most common in complex FC in their study.

Pokorn et al. conducted a prospective study at an institution in Slovenia, and the virus was detected in 140 (72.9%) of 192 cases of FC [29]. Whereas in the control groups who did not have FC, only 72 (51.4%) of 156 were virus positive. In their study, the most frequently detected viruses in patients with FC were IFV, HPIV, HRSV, and HCoV compared with the control group. The age-adjusted odds ratio was the highest for IFV (79.4) and rotavirus (22.0). Compared with other viruses, patients infected with norovirus had a shorter seizure duration and a lower risk of developing complex FC. Francis et al. conducted an observational study at a tertiary children’s hospital in Australia, and the virus was detected in 102 (71%) of 143 patients with FC [26]. The most frequently detected viruses were HRV (22%), HAdV (21%), enterovirus (20%), IFV (13%), and HHV6 (12%). More than one virus was found in 34% of FC patients. There was no significant clinical difference between children with confirmed pathogens and those without detected pathogens. In their study, 11% of the patients were vaccinated two weeks before the FC. In children with FC, respiratory viral Infections were more commonly found than the recent vaccination history. Our study did not analyze the parts of vaccination due to the limitations of the HIRA dataset.

In a retrospective study conducted in Hongkong that analyzed the association between pediatric FC and the virus, 923 FC-related admissions were analyzed, and 565 were first seizures [6]. Influenza, HAdV, HPIV, RSV, and rotavirus were identified in 17.6%, 6.8%, 6%, 2.7%, and 1.3%, respectively. However, 20.5% of patients relapsed, and there was no correlation with the virus type. In addition, the FC incidence for each virus infection was 20.8% for influenza, 18.4% for HPIV, 5.3% for HAdV, and 4.3% for RSV. The relative risk of FC was similarly high for influenza, HPIV and HAdV infection, and the relative risk of FC for RSV and rotavirus was low.

Han et al. analyzed the relationship between viral infections and FC using the HIRA data [23]. After extracting information from children between the ages of 1 month and 5 years diagnosed with FC and viral infection, they classified the patients according to virus-related diagnoses. Among all viral infections, the most common etiology was IFV (52.0%), followed by enterovirus (38.9%). Rotavirus (3.1%), HRSV (2.1%), and adenovirus (2.0%) were detected. The correlation between viruses and FC was significant in enterovirus infection but not in IFV infection. In our study, FC incidence showed a statistically significant association with increased HRSV, HCoV, norovirus, HRV, and HMPV infections. Only HRSV was associated with all age groups, and the results for the other viruses differed by age group. These results are like those of the previous studies.

However, the association between HRSV and FC remains controversial. In a study by Chung et al., the risk of developing FC was lower with HRSV or rotavirus than with IFV, HAdV, or HPIV [6]. In the study by Tebeila et al., HRSV infection was lower in patients with FC (4.1%) than in children without FC (11.0%) [30], and in the study by Cha et al., only 2.2% of patients with HRSV showed FC [31].

Rotavirus and norovirus have been reported as the cause of seizure associated with viral gastroenteritis. Rotavirus is known to be more often accompanied by fever than norovirus [32]. In the Taiwan study, 1444 children with norovirus-associated gastroenteritis were analyzed. Among them, 108 (7.48%) children caused seizure, the age of patients was 2.31 ± 2.12 years, and most patients (92.6%) were under 5 years old. 54.6% of patients experienced afebrile seizure [33]. Another study in Taiwan analyzed the molecular epidemiology of norovirus gastroenteritis with seizure, seizure occurred in 20.9% of norovirus infections, and GII.4 Den_Haag_2006b and GII.4 Sydney 2012 were the main variants correlated with convulsions [34]. They insisted that the rotavirus vaccination program changed the clinical pattern of rotavirus-related convulsions and effectively reduced febrile convulsions in children under 2 years of age. They expected that norovirus would be an important cause of febrile convulsions in children in the future. In our study, rotavirus was not associated with febrile convulsions, while norovirus was related to the occurrence of febrile convulsions.

Despite these findings, this study had several limitations. First, this was a retrospective study, which has a lower level of evidence than prospective studies and may have selection bias. Second, as a limitation of the data, the groups of patients diagnosed with FC and those infected with the virus differed. Third, no information is available on the degree of fever in relation to the type of virus and FC. Fourth, we cannot identify the vaccination history with the HIRA dataset alone. Finally, we cannot identify the vaccine-induced convulsions. Therefore, the statistically significant association between viral PDR and FC incidence cannot be interpreted as a direct relationship between the viruses and FC.

## 5. Conclusions

This study evaluated the incidence of FC and investigated its correlation with the prevalence of common viral respiratory and gastrointestinal infections in Korea. The incidence of FC without underlying diseases was 622.4/100,000 individuals. The HRSV infection was associated with the incidence of FC in all age groups. HCoV, norovirus, HRV, and HMPV were associated with FC incidence in certain age groups: HCoV and norovirus at 3–4.9 years of age and HRV and HMPV at 1–1.9 years of age. A large-scale prospective study is needed to ascertain the direct causality between FC and viral infections.

## Figures and Tables

**Figure 1 children-10-00529-f001:**
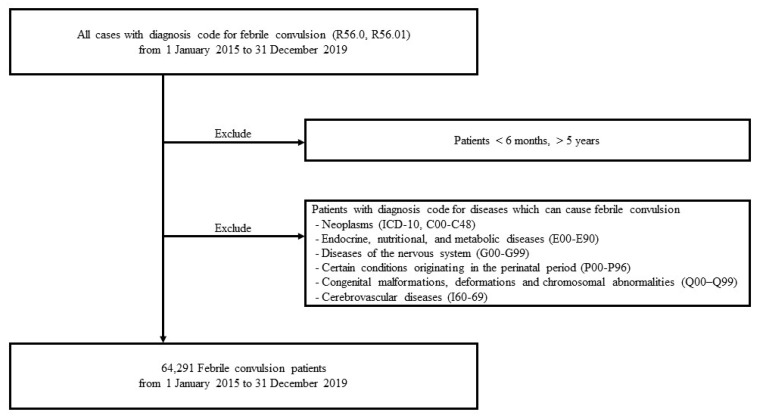
Flowchart illustrating patient selection.

**Figure 2 children-10-00529-f002:**
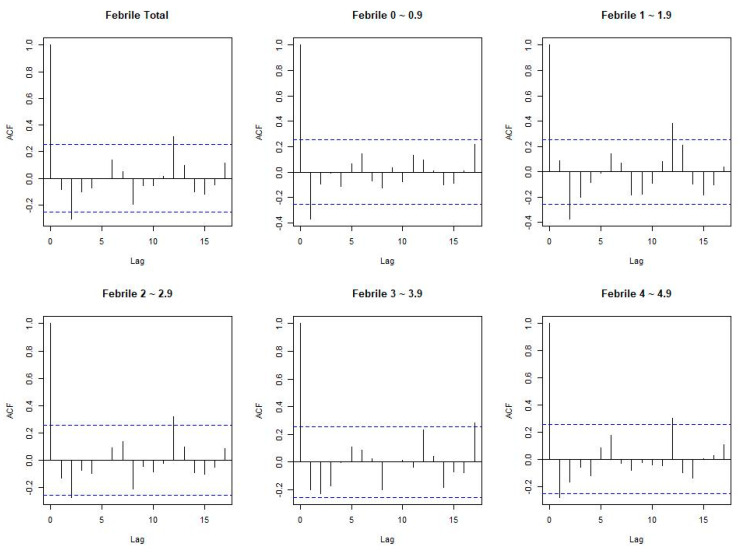
The autocorrelation function (ACF) is a statistical technique to determine the degree of correlation between the values in a time series. ACF after differencing.

**Figure 3 children-10-00529-f003:**
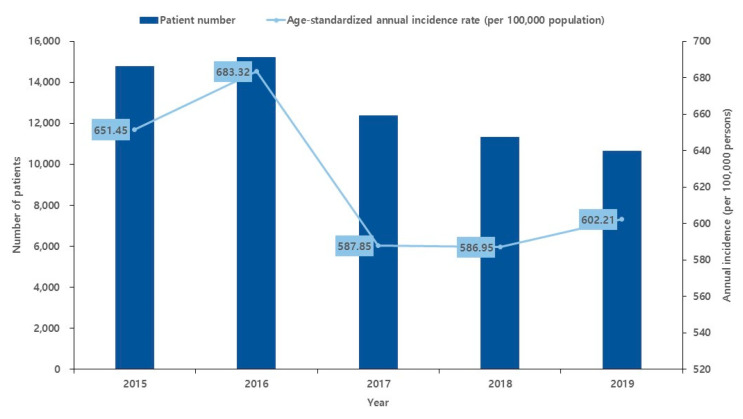
Annual incidence and age-standardized incidence rate of febrile convulsion from 2015 to 2019.

**Figure 4 children-10-00529-f004:**
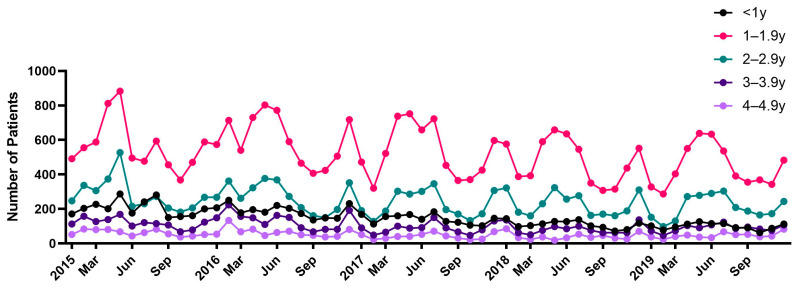
Monthly incidence trend of febrile convulsion according to age group from 2015 to 2019.

**Figure 5 children-10-00529-f005:**
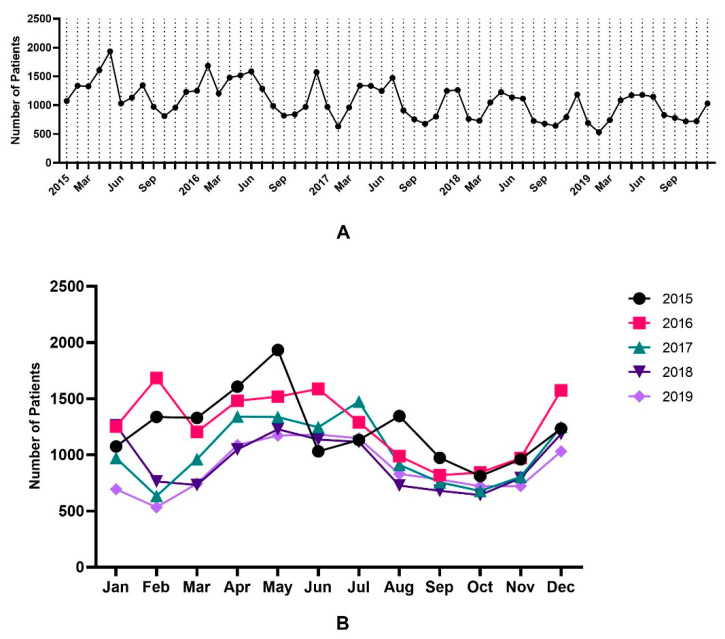
Annual incidence trend of febrile convulsion from 2015 to 2019; (**A**). Monthly trend analysis of febrile convulsion from 2015 to 2019; (**B**). Monthly trend analysis of febrile convulsion according to year.

**Figure 6 children-10-00529-f006:**
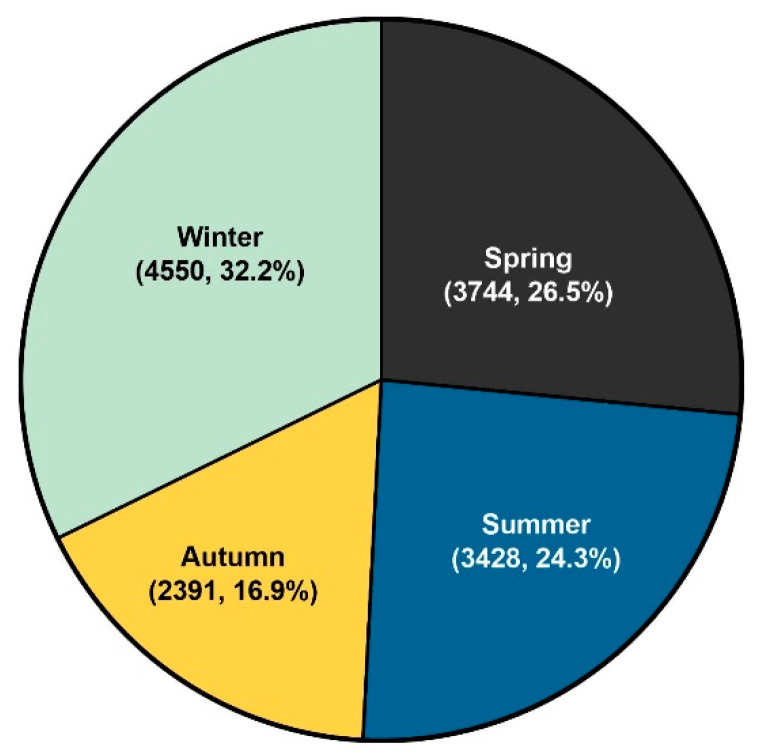
Seasonal trend of febrile convulsion incidence. Spring (March to May), summer (June to August), autumn (September to November) and winter (December to February). (Cumulative incidence for 5 years, %).

**Figure 7 children-10-00529-f007:**
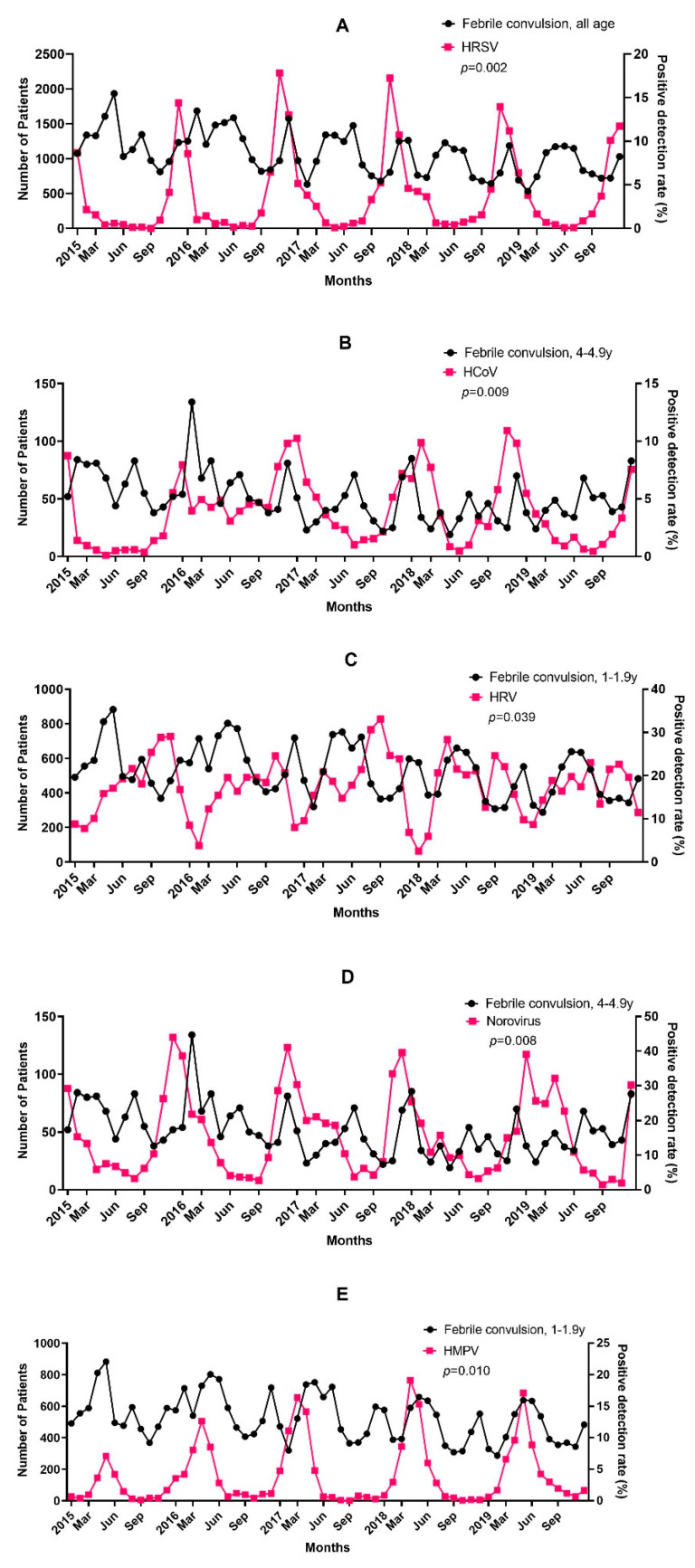
Relationship between PDR of (**A**) HRSV, (**B**) HCoV, (**C**) HRV, (**D**) HRSV, (**E**) HMPV and incidence of FC during the study period.

**Table 1 children-10-00529-t001:** Parameters of ARIMA models for febrile convulsion patients by age.

Parameters	0–0.9 Years	1–1.9 Years	2–2.9 Years	3–3.9 Years	4–4.9 Years	Total
p autoregressive	4	0	0	0	0	0
d difference	1	1	1	1	1	1
q moving average	1	0	0	0	2	0
AIC	583.98	748.16	695.48	603.99	528.67	841.17

**Table 2 children-10-00529-t002:** Characteristics of patients.

Variables		N (%)
Total number of patients		64291 (100.0)
Age (years)		1.55 ± 0.86
Age group (years)		
	0–0.99	8930 (13.8)
	1–1.99	31,485 (49.0)
	2–2.99	14,641 (22.8)
	3–3.99	6162 (9.6)
	4–4.99	3073 (4.8)
Sex		
	Male	34,278 (53.3)
	Female	30,013 (46.7)
Location		
	Seoul	11,367 (17.7)
	Pusan	3352 (5.2)
	Incheon	3531 (5.5)
	Daegu	2061 (3.2)
	Gwangju	2714 (4.2)
	Daejeon	1817 (2.8)
	Ulsan	2170 (3.4)
	Gyeonggi	16,015 (24.9)
	Gangwon	1859 (2.9)
	Chungbuk	2484 (3.9)
	Chungnam	2154 (3.4)
	Jeonbuk	2050 (3.2)
	Jeonnam	2888 (4.5)
	Gyeongbuk	2909 (4.5)
	Gyeongnam	5766 (9.0)
	Jeju	1061 (1.7)
	Sejong	88 (0.1)
Type of Insurance		
	Medical insurance	63,436 (98.7)
	Medical aid	855 (1.3)
	Free	-

**Table 3 children-10-00529-t003:** The age-standardized incidence rate of febrile convulsion by age group.

Age Group (Years)	Annual Incidence ^a^ (Annual Incidence Rate) ^b^					Overall Incidence Rate ^b^	Relative Risk ^c^
	2015	2016	2017	2018	2019		
0–0.9	2455 (578.2)	2271 (576.9)	1696 (490.5)	1319 (415.2)	1189 (402.9)	492.7	3.7
1–1.9	6778 (1547.5)	7243 (1639.7)	6396 (1560.7)	5749 (1589.8)	5319 (1607.1)	1589.0	11.9
2–2.9	3370 (765.9)	3302 (751.8)	2721 (614.3)	2744 (667.3)	2504 (690)	697.9	5.2
3–3.9	1421 (290.8)	1618 (367.3)	1048 (238.3)	1007 (227)	1068 (259.2)	276.5	2.1
4–4.9	743 (156.3)	777 (158.9)	500 (113.4)	494 (112.2)	559 (125.9)	133.3	1.0
Total	14767 (651.5)	15211 (683.3)	12361 (587.8)	11313 (587)	10639 (602.2)	622.4	

Data are presented as number (%); a. total number of patients by year; b. all rates are per 100,000 populations, directly age-adjusted to the 2010 population; c. relative risk compared with the 4–4.9 years age group.

**Table 4 children-10-00529-t004:** Analysis of the correlation between the incidence of febrile convulsion by age and the positive rate of the virus.

Age Group (Years)	HAdV	HPIV	HRSV	IFV	HCoV	HRV	HBoV	HMPV	Rotavirus	Norovirus	Adenovirus	Astrovirus
0–0.99	0.895	0.835	**0.007**	0.961	0.314	0.878	0.723	0.744	0.342	0.087	0.810	0.056
1–1.99	0.314	0.451	**0.012**	0.148	0.414	**0.039**	0.847	**0.010**	0.105	0.218	0.488	0.177
2–2.99	0.138	0.876	**0.005**	0.451	0.143	0.295	0.381	0.077	0.883	0.103	0.652	0.444
3–3.99	0.124	0.784	**0.000**	0.198	**0.022**	0.687	0.592	0.851	0.540	**0.038**	0.944	0.466
4–4.99	0.342	0.639	**0.003**	0.425	**0.009**	0.051	0.769	0.061	0.957	**0.008**	0.999	0.502
Total	0.201	0.905	**0.002**	0.252	0.143	0.290	0.627	0.089	0.630	0.091	0.698	0.235

Values of the Granger causality test between the time series of febrile convulsion diagnosis and the time points of positive detection rates of the virus, with <0.05 indicating significance (written in bold).

## Data Availability

The data presented in this study are available upon request from the corresponding author.

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
