# Peer review of "Seasonal Trend of Viral Prevalence and Incidence of Febrile Convulsion: A Korea Public Health Data Analysis"

_children, 2023, doi:10.3390/children10030529_

Round 1

Reviewer 1 Report

Thank you for the opportunity to review this paper byHa Rim Keum et al.

I think it is a well done investigation utilizing the important Korean database. It is good to see that these data are used and produce better insight in the important and frequent problem of febrile convulsions of children.

The study is well designed, performed and explained. May be the Authors should mention in the "Limitations" section yjat, given the study design, there is, obviously, no information available on the degree of fever in relation to type of virus and FC.  

I have several limitations too, as a clinician, interested in epidemiology of infections, my statistical knowledge is far from being able to review the autors approach, which seems soind and elegant to me.

Congratulations to the Authors 

Author Response

Thank you for reviewing our manuscript.

We correct as your recommendation.

Added content to Line 266-267.

Third, there is no information available on the degree of fever in relation to type of virus and FC.

Reviewer 2 Report

Overall, this paper is relatively well written and the data and statistical analyses appear to follow standard statistical protocols. The findings are reported accordingly with suitable statements of limitations of the study and findings within the context of other relevant studies.

The one recommendation I have for revision of the paper pertains to the statement in Section 2.3. Statistical Analysis:  "We then constructed a model of variations in FC diagnosis using the 93 autoregressive integrated moving average (ARIMA) modeling approach, which assumes 94 that current observations are related to past observations over time." Okay, that is good, but, given the critical importance of this analysis to the findings of the study, you then should include a table giving the corresponding results of this analysis (and the associated Granger analysis) and an associated paragraph(s) within the Results section that fully describes the findings. These analyses and findings are too central to the paper to be relegated to Supplementary materials. 

Author Response

Thank you for reviewing our manuscript.

We correct as your recommendation.

According to the reviewer's opinion, supplementary table 1 and supplementary figure 1 were modified to table 1 and figure 2, respectively, and added to the text.

Round 2

Reviewer 2 Report

Thank-you for adding Table 1 and Figure 2 to the text. But you also need to add corresponding text that describes and explains what the contents of the Table 1 and Figure 2 are and how they indicate good model fits, etc.

Author Response

In the opinion of our team's statistician, 2.3 statistical analysis section contains all the information necessary for statistical analysis. There is also information about Figures 2 and 1.

Nevertheless, we added an explanation of the statistical method according to the reviewer's recommendation. (lines 104-109)

The Granger causality test is a statistical technique that analyzes the causal relationship of each time series data, and each time series has an assumption that must follow stationarity. Therefore, it has a normalization process of time series data. At this time, ACF can know whether the time series data are correlated, and is used to identify the ARIMA model. Accordingly, the ACF in this study was shown in Figure 2, and Table 1 describes the model and diagnosis of ARIMA.